# The Protective Effect of a Dietary Extract of Mulberry (*Morus alba* L.) Leaves against a High Stocking Density, Copper and Trichlorfon in Crucian Carp (*Carassius auratus*)

**DOI:** 10.3390/ani13162652

**Published:** 2023-08-17

**Authors:** Gangfu Chen, Jiao Long, Huatao Li, Jing Xu, Jia Yuan, Qihui Yang, Lin Feng, Min Wu, Jun Jiang

**Affiliations:** 1Key Laboratory of Sichuan Province for Conservation and Utilization of Fishes Resources in the Upper Reaches of the Yangtze River, College of Life Sciences, Neijiang Normal University, Neijiang 641100, China; fanyahr@163.com (G.C.);; 2College of Fisheries, Guangdong Ocean University, Zhanjiang 524088, China; 3Animal Nutrition Institute, Sichuan Agricultural University, Chengdu 611130, China

**Keywords:** dietary supplement, bioenergetic homeostasis, growth performance, digestion capacity

## Abstract

**Simple Summary:**

Fish are maintained at high densities in modern fish farming systems, which explains their high susceptibility to infections. Copper sulfate and trichlorfon are commonly applied to fish for the control of infections and water quality. A high stocking density, Cu exposure, and trichlorfon exposure are typical inducers of physiological stress for fish. We found that adding the extract of mulberry (*Morus alba* L.) leaves (EML) to crucian carp (*Carassius auratus*) feed regulated their digestive capacity, antioxidant status, amino acid metabolism and bioenergetic homeostasis, thereby relieving the detrimental effects of these inducers on fish growth performance, feed intake and fish locomotion.

**Abstract:**

This study was designed to examine the protective effects of the extract of mulberry (*Morus alba* L.) leaves (EML) on crucian carp (*Carassius auratus*) against a high stocking density, Cu exposure and trichlorfon exposure, which adversely impact fish growth performance, feed intake and fish locomotion. High stocking densities decreased the activities of amylase, lipase, trypsin, Na^+^/K^+^-ATPase and alkaline phosphatase (AKP), and increased the content of malonaldehyde (MDA) in fish digestive organs, indicating an impairment of the digestive function and a disturbance of the antioxidant status. Cu exposure increased the activities of glutamate–oxaloacetate transaminase (GOT) and glutamate–pyruvate transaminase (GPT) in fish digestive organs, suggesting the activation of amino acid metabolism. Furthermore, trichlorfon exposure reduced the activities of lactate dehydrogenase (LDH), glutathione reductase (GR), GOT and GPT, and the capacities of the anti-superoxide anion (ASA) and anti-hydroxyl radical (AHR) in fish muscles, indicating a disruption of the bioenergetic homeostasis and antioxidant status. Our present study indicates that dietary EML supplementation relieved the detrimental effects induced by these stressors.

## 1. Introduction

Stocking density is directly related to fish productivity and is regarded as a determining factor of economic return [1]. Increasing the fish stocking density can improve fish yields without additional system costs [2]. However, crowded conditions cause significant detrimental effects, as evidenced by reduced food consumption and body weight gain in rainbow trout [3]. The growth performance of fish is associated with the digestion and absorption of nutrients [4]. Hence, there is the probability that crowded conditions have a detrimental influence on the digestion and absorption of nutrients so as to reduce the feed intake and growth rate. A high stocking density, which serves as a common strategy and procedure in fish farming, is a stressor for fish. In rainbow trout, crowding conditions increased oxidative stress [3]. These results imply that high stocking densities may degrade the digestive capacity of fish via disrupting the oxidant–antioxidant status of digestive organs, which needs further investigation.

High stocking densities can also undermine fish health [2]. Specifically, under crowded conditions, the susceptibility of fish to infections is increased. Copper sulfate is effective in improving water quality and controlling infections and is commonly applied to fish [5]. The hepatopancreas and intestine are important digestive and absorptive organs for fish [6]. For Jian carp (*Cyprinus carpio* var. Jian), Cu exposure caused significant detrimental effects, such as protein oxidation, lipid peroxidation, and alterations in the antioxidant activities of the digestive and absorptive organs [7]. In this sense, it is necessary to analyze the relationship between copper sulfate and the digestion capacity of aquatic animals.

High stocking densities increase the risk of infections induced by various pathogens, and infected fish are often treated with organophosphate compounds, such as trichlorfon [8]. In practice, trichlorfon is often used excessively, leading to toxic effects on fish, especially on their skeletal muscles [9], which play an important role in fish body movement. Furthermore, trichlorfon exposure induces oxidative stress in fish liver, which manifests as a decreased glutathione (GSH) content, a reduction in the activities of antioxidant enzymes, and intensified lipid peroxidation in common carp [10]. Systematic studies are required to further examine the potential correlation between oxidative stress in the muscles and the erratic movement of caused caused by trichlorfon.

Mulberry (*Morus alba* L.) leaves, rich in nutritional components, have been extensively used for various medicinal purposes since ancient times [11]. Our laboratory test reported that dietary supplementation with an extract of mulberry (*Morus alba* L.) leaves (EML) enhanced the growth performance of crucian carp (*Carassius auratus*) [12]. Similarly, studies on catfish (*Heteropneustes fossilis*) and carp (*Labeo rohita*) showed that the dietary supplementation of fermented mulberry leaves improved fish growth, as well as their digestive and absorptive capacities [13]. However, information concerning the effects of EML on fish kept at a high stocking density is limited. A previous study has demonstrated that EML supplementation reduced cadmium in the liver of rare minnow (*Gobiocypris rarus*) [14]. The effects of EML on the digestive and absorptive capacity of fish exposed to heavy metals, such as Cu, remain unclear. Furthermore, the dietary inclusion of mulberry leaves improved the hepatic antioxidant capacity, manifested as the increased activities of catalase (CAT) and total superoxide dismutase (T-SOD), and the elevated content of hepatic glutathione in juvenile *Megalobrama amblycephala* [15]. In pigs, dietary mulberry leaves (6%) showed a beneficial effect on muscle quality by modulating the expression of several key genes [16]. Despite these findings, the effects of EML on the antioxidant status of trichlorfon-treated fish muscles need further investigation.

The present study was directed to assess the potential protective effects of a dietetic EML supplementation on fish maintained at a high density and exposed to CuSO_4_ and trichlorfon. These effects might be related to the fish’s digestive and absorptive capacity, the balance of the antioxidant status, and bioenergetic homeostasis.

## 2. Materials and Methods

### 2.1. Chemical Reagent

Ethyl ether (AR), acetone (AR), cyclohexane (AR) and CuSO_4_·5H_2_O (AR) were purchased from Chengdu Kelong Chemical Reagent Factory (Chengdu, China). Trichlorfon (≥90%) was provided by Shanghai Biochemical Reagent Co., Ltd. (Shanghai, China). All other chemicals used in this experiment were of analytical reagent grade (AR).

### 2.2. Preparation of Extract of Mulberry (Morus alba *L.*) Leaves

Mulberry leaves were provided by a mulberry farmer in Neijiang (Sichuan, China) and the identification of botany was performed in Neijiang Normal University, where researchers assigned the voucher samples a reference number and subsequently deposited them. The procedures for the preparation of the mulberry (*Morus alba* L.) leaves extract were performed as previously described by our group [12]. The dry cyclohexane extract (CHE), ethyl ether extract (EEE), acetone extract (AE), and aqueous extract (AQE) were kept in a dark place in sealed bottles and preserved at −80 °C before use.

### 2.3. Determination of Flavonoid Content

The flavonoid content of the mulberry (*Morus alba* L.) leaves extract was determined according to the methods of Jia et al. [17].

### 2.4. The Composition Analyses of Ethyl Ether Extract of Mulberry (Morus alba *L.*) Leaves

The ethyl ether extract of the mulberry (*Morus alba* L.) leaves was dissolved in ethyl acetate to prepare the saturated solution. Samples were analyzed using a gas chromatograph-mass spectrometer (GC/MS) system according to the methods of Li et al. [18], with a slight modification. The GC/MS (Angilent 7890, Wilmington, DE, USA) was equipped with a flame ionization detector (FID) and a HP-5MS column (30 × 0.25 mm, 0.25 μm). Nitrogen, used as the carrier gas, remained stable at 1 mL/min. The injection temperature and FID temperature were kept at 250 °C. The column temperature was programmed firstly at 40 °C for 1.0 min, increasing to 220 °C at 4.0 °C/min and to 250 °C at 10.0 °C/min for 3 min. The composition analyses are shown in Figure 1 and Table 1.

### 2.5. Experimental Fish and Diets

Juvenile crucian carps (*Carassius auratus*) were purchased from a fish farm in Neijiang (Sichuan, China). Fish were maintained under lab conditions (22.0 ± 1 °C) with a natural light–dark cycle. The procedures for the diet preparation were performed as previously conducted by our group [12]. The composition and nutrients content of the basal diet are given in Table 2. The basal diet contained a protein content of 34.73% and a lipid content of 5.56%. The experimental diets were supplemented with the mulberry (*Morus alba* L.) leaf ethyl ether extract at 0.0, 1.0, 2.0, 3.0, 4.0 5.0, 6.0, and 7.0 g/kg of diet.

### 2.6. Protection Assays for Dietary Ethyl Ether Extract of Mulberry (Morus alba *L.*) Leaves on High-Density Conditions

The procedures for the density assays were performed as previously described by our group [19]. Juvenile crucian carp (7.2 ± 0.2 g) were randomly assigned into nine groups, each of which was contained in 4 replicate aquariums. Group one (control), which contained 15 fish per aquarium (0.48 fish/L), was the optimum density group. The other groups, with 30 fish per aquarium (0.97 fish/L), were the high-density groups. The control group was fed with the Basal diet, and the experimental diets in groups two to nine were supplemented with the mulberry (*Morus alba* L.) leaf ethyl ether extract at 0.0, 1.0, 2.0, 3.0, 4.0 5.0, 6.0, and 7.0 g/kg for 60 days. Half an hour after feeding, the uneaten feed was captured via siphoning, then dried and reweighed to calculate the feed intake (FI). At the beginning and end of the experiment, fish were weighed and counted for the following calculation.
Survival rate (SR) (%) = 100 × (final number/initial fish number)
Feed intake (FI) (g/fish) = feed consumed (g)/final fish number
Weight gain (WG) (g/fish) = harvest weight (g/fish) − initial weight (g/fish)
Feed efficiency (FE) (%) = 100 × WG/FI
Specific growth rate (SGR) (%/day) = 100 × (ln harvest weight − ln initial weight)/experimental duration day

Sample collection procedures were performed as previously conducted by our group [12]. At the end of the trial, the fish were anesthetized using benzocaine (50 mg/L), and then the intestine and hepatopancreas were immediately collected and stored at −80 °C for analysis. The content of protein, trypsin, lipase and amylase activities, as well as the Na^+^/K^+^-ATPase and alkaline phosphatase (AKP) activities, were measured. The content of malonaldehyde (MDA), glutathione peroxidase (GPx), catalase (CAT) and superoxide dismutase (SOD) activities, as well as the capacities of the anti-hydroxyl radical (AHR) and anti-superoxide anion (ASA), were also determined.

### 2.7. Protection Assays of Dietary Ethyl Ether Extract of Mulberry (Morus alba *L.*) Leaves on CuSO_4_ Exposure

The CuSO_4_ exposure was determined according *to* the general Organization for Economic Cooperation and Development (OECD) guidelines, following our previous report [19,20]. A total of 480 juvenile crucian carp (10.5 ± 0.3 g) were randomly assigned into eight groups, each of which was contained in 4 replicate aquariums (15 fish in each aquarium). The fish in the 8 treatment groups were fed with diets that added the mulberry (*Morus alba* L.) leaf ethyl ether extract at 0.0, 1.0, 2.0, 3.0, 4.0 5.0, 6.0, and 7.0 g/kg for 30 days, respectively. After that, 30 fish from each group were divided into 8 groups at random, each of which contained 10 fish; these were maintained under the concentration of Cu at 0.7 mg/L for four days. An additional 30 fish in the control group experienced exposure to the clean water. After that, the FI in each treatment group was confirmed and the fish were anaesthetized, as is standard when studying the protective effect of the dietary ethyl ether extract of mulberry (*Morus alba* L.) leaves on fish kept at a high density. Then, the intestine and hepatopancreas were collected and stored at −80 °C for the analysis of trypsin, amylase, lipase, and alkaline phosphatase, as well as the analysis of glutamate–pyruvate transaminase (GPT) and glutamate–oxaloacetate transaminase (GOT) activities.

### 2.8. Protection Assays of Dietary Ethyl Ether Extract of Mulberry (Morus alba *L.*) Leaves on Trichlorfon Exposure

The procedures used to assess trichlorfon exposure were performed as previously conducted by our group [21]. The feeding trial was the same as that used when studying the protective effects of the dietary mulberry (*Morus alba* L.) leaf ethyl ether extract on CuSO_4_ exposure.

After the feeding trial, 30 fish from each group, each of which contained 10 fish, were maintained under 2.2 mg trichlorfon/L water for four days. Another 30 fish (negative group) were maintained under clean water, each of which contained 10 fish. The rollover rates of each group were recorded. At the end of the trial, the fish were anaesthetized, and the dorsal fillets of the fish were immediately collected after skinning and stored at −80 °C for analysis. The content of protein and protein carbonyl (PC), the anti-superoxide anion and anti-hydroxyl radical capacity, as well as the glutamate–oxaloacetate transaminase, glutamate–pyruvate transaminase, glutathione reductase (GR), lactate dehydrogenase (LDH) and catalase activities were determined.

### 2.9. Biochemical Analysis

The activities of lipase, trypsin, Na^+^/K^+^-ATPase, amylase, alkaline phosphatase, glutamate–oxaloacetate transaminase and glutamate–pyruvate transaminase were determined according to our previous study [4]. The capacity of the anti-hydroxyl radical and anti-superoxide anion, the level of protein carbonyl and malonaldehyde, as well as the activities of lactate dehydrogenase, glutathione reductase, superoxide dismutase, catalase and glutathione peroxidase were determined as described by Jiang et al. [7].

### 2.10. Statistical Analysis

All data (mean ± SD) were subjected to a one-way or two-way ANOVA by using SPSS 13.0. Orthogonal polynomial contrasts were performed to determine whether the effect was linear and/or quadratic. Duncan’s multiple range test was performed to compare the treatment means.

## 3. Results

### 3.1. Flavonoid Content in Extract of Mulberry (Morus alba *L.*) Leaves

The flavonoid content the mulberry (*Morus alba* L.) leaf extract is shown in Table 3. The mulberry (*Morus alba* L.) leaf ethyl ether extract contained a higher flavonoid content than the other extracts (*p* < 0.05), followed by the mulberry (*Morus alba* L.) leaf acetone extract. The flavonoid contents were lower in the cyclohexane extract and aqueous extract.

### 3.2. Effects of Dietary Ethyl Ether Extract of Mulberry (Morus alba *L.*) Leaves on Fish Growth Performance under High-Density Conditions

As shown in Table 4, the fish survival rate (SR) was not influenced (100%) by the high-density treatment (*p* > 0.05). However, their feed intake, weight gain, feed efficiency and specific growth rate were significantly reduced (*p* < 0.05). Their final body weight and weight gain were significantly enhanced with the increase in the dietary intake levels of the mulberry (*Morus alba* L.) leaf ethyl ether extract up to 4.0 g/kg (*p* < 0.05), and plateaued thereafter with a further increase in the dietary content of the mulberry (*Morus alba* L.) leaf ethyl ether extract (*p* > 0.05). Similar trends were found in the specific growth rate, feed intake and feed efficiency, which improved with the increase in the dietary levels of the mulberry (*Morus alba* L.) leaf ethyl ether extract up to 5.0, 5.0 and 3.0 g/kg, respectively. Based on the broken-line analysis, the optimum dietary supplementation of the mulberry (*Morus alba* L.) leaf ethyl ether extract was 4.85 g kg/diet, estimated using the recovery rate of weight gain (RWG) for crucian carp under high stocking densities (Figure 2).

### 3.3. Effects of Dietary Mulberry (Morus alba *L.*) Leaf Ethyl Ether Extract on Fish Digestive and Absorptive Enzyme Activities, as Well as Antioxidant Capacity, under High-Density Conditions

As shown in Table 5, the high-density treatment led to a slight decrease in the trypsin activity of the crucian carp hepatopancreas. The activities of lipase in the hepatopancreas, as well as those of lipase, amylase, Na^+^/K^+^-ATPase and alkaline phosphatase in the intestine, were significantly reduced under the high-density treatment (*p* < 0.05). The activities of trypsin and lipase in the hepatopancreas were significantly enhanced with the increase in the dietary levels of the mulberry (*Morus alba* L.) leaf ethyl ether extract up to 5.0 and 2.0 g/kg (*p* < 0.05), respectively, and plateaued thereafter with a further increase in the dietary content of the mulberry (*Morus alba* L.) leaf ethyl ether extract (*p* > 0.05). Similar trends were found in the lipase and Na^+^/K^+^-ATPase activities of the intestine, which improved with the increase in the dietary levels of the mulberry (*Morus alba* L.) leaf ethyl ether extract up to 4.0 and 3.0 g/kg, respectively. The activities of amylase in the intestine of the fish fed diets comprising 5.0 g/kg of the mulberry (*Morus alba* L.) leaf ethyl ether extract were higher than other levels. The activities of alkaline phosphatase in the intestine slightly increased with the dietary content of the mulberry (*Morus alba* L.) leaf ethyl ether extract under the high-density treatment.

As shown in Table 5, the high-density treatment led to an increase in the malonaldehyde levels in the hepatopancreas and intestine of the crucian carp (*p* < 0.05). However, dietary supplementation with the mulberry (*Morus alba* L.) leaf ethyl ether extract decreased these parameters. The activities of catalase in the hepatopancreas and the capacity of the anti-hydroxyl radical in the intestine were significantly reduced under the high-density treatment (*p* < 0.05). The activities of catalase in the hepatopancreas and the capacity of the anti-hydroxyl radical in the intestine were significantly enhanced with the increase in the dietary levels of the mulberry (*Morus alba* L.) leaf ethyl ether extract up to 5.0 and 4.0 g/kg (*p* < 0.05), respectively, and plateaued thereafter with a further increase in the dietary content of the mulberry (*Morus alba* L.) leaf ethyl ether extract (*p* > 0.05). The high-density treatment led to a slight decrease in the glutathione peroxidase activity in the hepatopancreas and intestine of the crucian carp and a slight increase in the superoxide dismutase activity in the intestine of the crucian carp. The activities of glutathione peroxidase in the hepatopancreas and intestine of the fish fed diets comprising 4.0 and 5.0 g/kg of the mulberry (*Morus alba* L.) leaf ethyl ether extract were higher than other levels, respectively (*p* < 0.05). The activities of superoxide dismutase in the intestine were not significantly influenced by the dietary content of the mulberry (*Morus alba* L.) leaf ethyl ether extract under the high-density treatment.

### 3.4. Effects of Dietary Mulberry (Morus alba *L.*) Leaf Ethyl Ether Extract on FI under Cu Exposure in Fish

As shown in Table 6, the feed intake (FI) was significantly reduced under Cu exposure (*p* < 0.05). The FI were significantly enhanced with the increase in the dietary levels of the mulberry (*Morus alba* L.) leaf ethyl ether extract up to 5.0 g/kg (*p* < 0.05), and plateaued thereafter with a further increase in the dietary content of the mulberry (*Morus alba* L.) leaf ethyl ether extract (*p* > 0.05). Based on the broken-line analysis, the optimum dietary supplementation of the mulberry (*Morus alba* L.) leaf ethyl ether extract was 4.52 g kg/diet, estimated using the feed intake of the crucian carp under Cu exposure (Figure 3).

### 3.5. Effects of Dietary Ethyl Ether Extract of Mulberry (Morus alba *L.*) Leaves on Digestive, Absorptive and Metabolic Parameters under Cu Exposure in Fish

As shown in Table 7, Cu exposure led to a slight decrease in the lipase activity in the hepatopancreas of the crucian carp. The activities of amylase in the hepatopancreas, as well as those of trypsin, amylase and alkaline phosphatase in the intestine, were significantly reduced, and the activities of glutamate–pyruvate transaminase and glutamate–oxaloacetate transaminase in the hepatopancreas were significantly increased under Cu exposure (*p* < 0.05). The activities of lipase in the hepatopancreas and trypsin in the intestine were significantly enhanced with the increase in the dietary levels of the mulberry (*Morus alba* L.) leaf ethyl ether extract up to 3.0 g/kg. The activities of amylase in the hepatopancreas and intestine were significantly enhanced with the increase in the dietary levels of the mulberry (*Morus alba* L.) leaf ethyl ether extract up to 3.0 and 4.0 g/kg (*p* < 0.05), respectively. The activities of alkaline phosphatase were significantly enhanced with the increase in the dietary levels of the mulberry (*Morus alba* L.) leaf ethyl ether extract up to 4.0 g/kg (*p* < 0.05), and plateaued thereafter with a further increase in the dietary content of the mulberry (*Morus alba* L.) leaf ethyl ether extract. The activities of glutamate–pyruvate transaminase and glutamate–oxaloacetate transaminase decreased with the increase in the dietary levels of the mulberry (*Morus alba* L.) leaf ethyl ether extract up to 5.0 g/kg.

### 3.6. Effects of Dietary Ethyl Ether Extract of Mulberry (Morus alba *L.*) Leaves on Rollover under Trichlorfon Stress in Fish

As shown in Table 8, rollover was significantly increased under trichlorfon stress (*p* < 0.05). Rollover was significantly reduced with the increase in the dietary levels of the mulberry (*Morus alba* L.) leaf ethyl ether extract up to 5.0 g/kg (*p* < 0.05), and plateaued thereafter with a further increase in the dietary content of the mulberry (*Morus alba* L.) leaf ethyl ether extract (*p* > 0.05). Based on the broken-line analysis, the optimum dietary supplementation of the mulberry (*Morus alba* L.) leaf ethyl ether extract was 4.90 g kg/diet, estimated using the inhibitory rate of rollover (IR) for crucian carp under trichlorfon stress (Figure 4).

### 3.7. Effects of Dietary Ethyl Ether Extract of Mulberry (Morus alba *L.*) Leaves on Metabolic Parameters and Antioxidant Status under Trichlorfon Stress in Fish Muscle

As shown in Table 9, trichlorfon stress led to a slight decrease in the anti-superoxide anion capacity in the crucian carp muscle. The activities of lactate dehydrogenase, glutamate–pyruvate transaminase, glutamate–oxaloacetate transaminase, catalase and glutathione reductase, as well as the capacity of the anti-hydroxyl radical, were significantly reduced under trichlorfon stress (*p* < 0.05). However, the dietary supplementation of the mulberry (*Morus alba* L.) leaf ethyl ether extract increased these parameters. The activities of lactate dehydrogenase, glutamate–pyruvate transaminase, glutamate–oxaloacetate transaminase and catalase, as well as the capacity of the anti-hydroxyl radical in the muscle of fish fed diets comprising 3.0, 4.0, 2.0, 3.0 and 7.0 g/kg of the mulberry (*Morus alba* L.) leaf ethyl ether extract, were higher than other levels, respectively. The activities of glutathione reductase and the capacity of the anti-hydroxyl radical were significantly enhanced with the increase in the dietary levels of the mulberry (*Morus alba* L.) leaf ethyl ether extract up to 4.0 g/kg (*p* < 0.05), and plateaued thereafter with a further increase in the dietary content of the mulberry (*Morus alba* L.) leaf ethyl ether extract (*p* > 0.05). The levels of protein carbonyl were significantly increased under trichlorfon stress (*p* < 0.05), and the dietary supplementation of the mulberry (*Morus alba* L.) leaf ethyl ether extract decreased these parameters.

## 4. Discussion

### 4.1. Dietary Extract of Mulberry (Morus alba *L.*) Leaves Relieves the Detrimental Effects of High-Density Conditions and Cu Exposure on Fish Growth Performance

Stocking density is identified as a critical part of aquaculture. The optimum stocking density is species-specific and depends on the fish size and water exchange rate [22]. For higher production and economic return, fish are maintained at high densities in modern fish farming systems [1]. In the present study, growth performance parameters, including the final body weight, weight gain, specific growth rate, feed intake and feed efficiency, decreased at a high stocking density, which indicates that fish growth was depressed by this inappropriately high density. A similar observation has been reported in rainbow trout, with high stocking densities having a negative effect on fish feed intake and growth performance [3]. Dietary EML supplementation increased the values of these parameters, which suggests that it conferred protection against the detrimental effect caused by the excessively high stocking density. These results conform to the findings of our previous laboratory report demonstrating that the dietary supplementation of mulberry (*Morus alba* L.) leaf extracts (EML) enhances the growth performance of crucian carp [12]. In the present study, broken-line analysis was performed to determine the optimum amount of dietary EML supplementation under the high stocking density, and based on the recovery rate of weight gain (RWG), the value was estimated to be 4.85 g kg/diet.

In the fish farming industry, the water quality and infections are commonly controlled by copper sulfate [5]. On the one hand, Cu is an essential metal for fish. On the other hand, as a heavy metal, its accumulation threatens aquatic ecosystems and humans [23]. In the present study, copper had significant detrimental effects on crucian carp, as indicated by their reduced feed intake. Consistent with our present results, a study on Indian Major Carp (*Cirrhinus mrigala*) showed that the feed intake reduced significantly in all Cu treatment groups [24]. In addition, copper increased crucian carp mortality [19]. Similar observations were obtained in *Cirrhinus mrigala* [24] and Caspian Sea kutum (*Rutilus frisii kutum*) [23]. In the present study, the feed intake was significantly decreased by CuSO_4_, and was increased by dietary EML supplementation, indicating that dietary EML alleviated the detrimental effect of Cu stress. In the broken-line analysis, the optimum amount of dietary EML supplementation under Cu exposure was 4.52 g kg/diet based on the rate of feed intake (RFI) for crucian carp.

### 4.2. Dietary Extract of Mulberry (Morus alba *L.*) Leaves Relieves the Detrimental Effect of Fish Digestive and Absorptive Capacity under High-Density Conditions and Cu Exposure

The growth performance of fish is associated with their nutrient digestion and absorption, which depend on the activities of digestive enzymes and brush border membrane enzymes [4]. Numerous fish digestive enzymes were synthesized in the exocrine pancreas and then secreted into the intestinal tract [25]. Such enzymes include trypsin, lipase and amylase [26]. In this study, their activities at a high stocking density indicated that the digestive capacity of fish was depressed by the excessively high stocking density. Consistent evidence can be seen in tilapia (*Oreochromis niloticus*) [27], Liao river shrimp (*Palaemonetes sinensis*) [28] and largemouth bass (*Micropterus salmoides*) [29]. The decrease in the activities of fish digestive enzymes might be due to the disrupted endocrine system and elevated cortisol levels induced by high stocking densities [29]. Alkaline phosphatase (AKP) and Na^+^/K^+^-ATPase are important enzymes responsible for the absorption of food molecules, and are commonly used as indicators for the absorption capacity of fish [4]. In the present study, their activities were reduced at a high stocking density, indicating the depressed absorptive capacity of crucian carp under this inappropriately high stocking density. In rainbow trout, high stocking densities have been found to decrease the expression of genes related to the function and integrity of the intestinal epithelium [30]. Our study showed that the activities of trypsin, lipase, amylase and alkaline phosphatase in the digestive organs decreased under CuSO_4_ exposure, which suggests that the digestive and adsorptive capacities of crucian carp were depressed when they were treated with CuSO_4_. Similar observations have been reported in juvenile *Epinephelus coioides*, with Cu exposure being found to hinder the activities of digestive enzymes, including protease, lipase and amylase in the liver and intestine [31]. The decrease in the activities of fish digestive enzymes possibly arose from the deposition of metal granules in the fish liver and the provocation of hepatic DNA damage [32]. In the present study, dietary EML supplementation increased the activities of trypsin, amylase, lipase, Na^+^/K^+^-ATPase and alkaline phosphatase, which indicates that it protects against the negative effects exerted by the inappropriately high stocking density and Cu exposure on the digestive and absorptive capacities of fish. These results conformed to our laboratory report finding that dietary EML supplementation improved the digestive and absorptive capacities of crucian carp [12]. To the best of our knowledge, our study was the first to propose that EML relieved the detrimental effects on the digestive capacity of fish under high stocking and Cu stress.

In fish, glutamate–pyruvate transaminase (GPT) and glutamate–oxaloacetate transaminase (GOT) are two vital enzymes involved in amino acid metabolism, and facilitate the utilization of amino acids as sources of energy via deamination [4]. Furthermore, the main end product of protein and amino acid metabolism is ammonia in teleost fish [33]. The present study showed that GOT and GPT activities increased under Cu exposure, which indicates that the protein and amino acid metabolism were activated by Cu stress. Parallel findings were documented in sculpin (*Myoxocephalus octodecemspinosus*), with Cu exposure elevating plasma ammonia [33]. In teleost, the increase in plasma ammonia is most likely due to the increased protein and amino acid breakdown rate arising from elevated cortisol levels under Cu exposure [34]. In the present study, dietary EML supplementation decreased GOT and GPT activities under Cu exposure, which indicates that it weakened the detrimental effect of Cu exposure on the amino acid metabolism of fish. Notably, no other reports focused on the relationship between EML and amino acid metabolism in fish under Cu stress. Further studies are needed to reveal the underlying mechanisms.

### 4.3. Dietary Extract of Mulberry (Morus alba *L.*) Leaves Relieves the Detrimental Effect of Oxidative Stress on Fish Digestive Organs under High-Density Conditions

In aquatic animals, the structure and function of their tissues and organs are largely dependent on the antioxidant status [35]. Reactive oxygen species (ROS) can be generated by aerobic metabolism or introduced by exogenous sources. They are detrimental to lipids and proteins [36]. Lipid peroxidation, with malonaldehyde (MDA) as a sensitive marker, occurs due to the oxidative deterioration of polyunsaturated fatty acids (PUFA), which are widely distributed in fish bodies [35]. In the present study, the MDA content was significantly higher in the hepatopancreas and intestine of crucian carp reared at a high stocking density, which indicates that the inappropriately high stocking density caused lipid peroxidation in their digestive organs. The study on largemouth bass (*Micropterus salmoides*) showed a consistent result, with high stocking densities negatively effecting the liver MDA content [29]. In the present study, the MDA content significantly increased at a high stocking density, and decreased after EML supplementation, which shows that dietary EML diminishes the negative influence of an excessively high stocking density on lipid peroxidation. These results conformed to our laboratory report stating that the increase in the MDA content induced by hypoxia/reoxygenation in the intestine and hepatopancreas was suppressed by dietary EML supplementation in crucian carp [12]. Flavonoids, an active ingredient of mulberry (*Morus alba* L.) leaf extract, inhibited MDA formation in rat liver [37]. They might have a favorable effect on preventing lipid peroxidation in the digestive organs of fish, which needs further investigation.

The increase in ROS, namely superoxide radicals (O_2_^•−^) and hydroxyl radicals (^•^OH), can induce lipid peroxidation [7]. The anti-hydroxyl radical capacity (indicative of ^•^OH-scavenging capacity) significantly decreased at a high stocking density, and increased after dietary EML supplementation, which indicates that dietary EML reversed the reduction in the ^•^OH-scavenging capacity induced by the inappropriately high stocking density. These results were supported by our laboratory report stating that dietary EML improved the ^•^OH-scavenging capacity in crucian carp digestive organs [12]. Our study is the first to examine the role of EML in mitigating the detrimental effects of excessive ^•^OH free radical production on the digestive organs of fish under high-density conditions.

Fish antioxidant defense systems, consisting of antioxidant enzymes, have evolved to avoid or repair the damage caused to fish tissues and organisms by ROS [38]. In these systems, superoxide dismutase is the initial responder to O_2_^•−^ and catalyzes it to H_2_O_2_ and the dioxygen molecules [35]. Then, H_2_O_2_ can be detoxified by CAT and GPx [29]. The activities of CAT decreased in the hepatopancreas under a high stocking density, implying that inappropriately high stocking densities have adverse effects on the antioxidant defense systems of the digestive organs of fish. Comparable findings were recorded in tilapia (Oreochromis niloticus) [27], juvenile Chinese sturgeon (*Acipenser sinensis*) [39] and largemouth bass (*Micropterus salmoides*) [29]. Dietary EML supplementation increased CAT and GPx activities in the digestive organs of fish under high stocking densities. These results matched our laboratory outcome, with dietary EML enhancing the antioxidant enzymes’ defense systems in the intestine and hepatopancreas of the crucian carp [12]. Quercetin, a kind of flavonol glycoside in mulberry leaves, reduced oxidative stress in the liver of mice [40]. We anticipate a similar effect in the case of the digestive organs of fish, which needs further investigation.

### 4.4. Dietary Mulberry (Morus alba *L.*) Leaf Extract Relieves the Detrimental Effect on Fish Muscle Function under Trichlorfon Stress

Trichlorfon makes the fish lose their balance by inhibiting acetylcholinesterase (AChE) activity at the neuromuscular junctions of fish muscle [9]. In the present study, trichlorfon induced significant detrimental effects, as indicated by crucian carp rollover. These phenomena were relieved by dietary EML supplementation, which reveals that dietary EML alleviated the detrimental effects of trichlorfon. In the broken-line analysis in the present study, the optimum dietary EML supplementation was estimated to be 4.90 g kg/diet based on the inhibitory rate of rollover (IR). Fish movements rely heavily on the energy metabolism in swimming muscles [41].

Lactate dehydrogenase (LDH), a general biomarker of stress, is responsible for the reversible transformation of pyruvate to lactate. It is a crucial enzyme for energy production in the anaerobic pathways [42]. In the present study, LDH activity was significantly lower in crucian carp muscles exposed to trichlorfon, contributing to an energetic imbalance. The muscle cytosol and mitochondria of silver catfish (*Rhamdia quelen*) showed a similar pattern [43]. It is apparent that trichlorfon exposure altered the bioenergetic homeostasis by inhibiting the activity of creatine kinase, which is an important enzyme for energy homeostasis and the activation of complexes II–III and IV of the respiratory chain (a crucial pathway for ATP production) [43].

GOT and GPT activities were also significantly lower in crucian carp muscles exposed to trichlorfon, contributing to an energetic imbalance. Fish utilize proteins for energy supply in normal conditions [44]. Under stressful conditions, such as exposure to pesticides, lipids are used as alternative energy sources in fish muscles, corroborating the alteration in the fatty acid profiles of silver catfish in the presence of trichlorfon [43]. In this study, the activities of GPT, LDH and GOT decreased in crucian carp muscles under trichlorfon exposure, and increased after dietary EML supplementation, which demonstrates that dietary EML protects against the inhibitory effect of trichlorfon on fish bioenergetic homeostasis. This may partly explain the trichlorfon-induced rollover of crucian carp. Nevertheless, the mechanism by which EML relieved the trichlorfon-induced muscle dysfunction in fish needs further exploration.

### 4.5. Dietary Mulberry (Morus alba *L.*) Leaf Extract Relieves the Detrimental Effect of Oxidative Stress on Fish Muscles under Trichlorfon Stress

Trichlorfon exposure increased the number of ROS in fish, resulting in protein oxidation and lipid peroxidation [8]. Protein oxidation is defined as the oxidative deterioration of protein, and the level of protein carbonyls is a marker of the oxidation degree [35]. Our present study showed that the contents of protein carbonyl were significantly higher in crucian carp muscles under trichlorfon exposure, contributing to protein oxidation. Similarly, it has been reported that trichlorfon-induced lipid oxidative damage impairs the gill function and cell structure of common carp (*Cyprinus carpio* L.) [45]. In the present study, dietary EML supplementation decreased the values of these parameters, indicating that it weakened the trichlorfon-induced damage.

Damage to the protein and lipid structure was correlated with the excessive production of free radicals, such as the superoxide and hydrogen peroxide [45]. In the present study, the anti-superoxide anion capacity (indicative of O_2_^•−^-scavenging capacity) and anti-hydroxyl radical capacity decreased under trichlorfon exposure, and was elevated after dietary EML supplementation, which indicates that dietary EML protects against the detrimental effect of excessive free radical production caused by trichlorfon. It was previously found that flavonoids exert antioxidant properties by scavenging superoxide anions in rats [37]. We anticipate that flavonoids might help scavenge free radicals in the case of fish muscles. Our study was the first to reveal that EML relieved the detrimental effect of excessive free radical production in fish muscles under trichlorfon exposure.

Fish antioxidant defense systems, consisting of enzymatic and non-enzymatic antioxidants, are crucial for protection against oxidative stress [38]. In enzymatic antioxidant enzyme systems, GSH S-transferase is a group of multifunctional enzymes that catalyze the conjugation of electrophilic metabolites to the thiol group of GSH [45]. At the expense of the NADPH, glutathione reductase (GR) catalyzes the reduction of glutathione (GSSG) to glutathione [46]. In this study, CAT and GR activities were significantly lower in crucian carp muscles exposed to trichlorfon, which indicates that the fish enzymatic antioxidant defense systems were disturbed by trichlorfon exposure. Similarly, it has been reported that trichlorfon exposure decreases CAT activities in the gills and livers of common carp (*Cyprinus carpio* L.) [45]. The decrease in antioxidant enzyme activities may be due to the excessive free radical production induced by trichlorfon. In the present study, dietary EML supplementation increased the activities of these enzymes, suggesting that it protects against the detrimental effect of the trichlorfon-induced imbalance between oxidants and antioxidants. Flavonoids exerted antioxidant properties by preventing GSH depletion in human red blood cells [47]. It is hypothesized that flavonoids can improve the antioxidant defense systems in fish muscles. This study represents a pioneering work to show that EML protected against the inhibitory effect of imbalance on fish muscles under trichlorfon exposure.

## 5. Conclusions

Under the inappropriately high stocking density, fish growth was depressed, their digestive and absorptive capacities were degraded, and their antioxidant status was interrupted. These parameters were improved after the dietary EML supplementation, indicating that it relieved the detrimental effect of a high stocking density on fish growth performance. In the broken-line analysis, the optimum amount of dietary EML supplementation was estimated as 4.85 g kg/diet according to the RWG of crucian carp under the high stocking density. The Cu exposure reduced the feed intake of the fish, and EML mitigated the disruptive effect of Cu stress on their digestive and absorptive capacities, as well as their antioxidant status. The optimum amount of dietary EML supplementation was 4.52 g kg/diet for crucian carp under Cu stress. Additionally, trichlorfon exposure induced detrimental effects, as indicated by the rollover of crucian carp. Specifically, the bioenergetic homeostasis and antioxidant status in fish muscles were interrupted by trichlorfon. This negative influence was mitigated after dietary EML supplementation at the optimum content of 4.90 g kg/diet, estimated using the IR. This mitigation could be attributed to the desirable effects of EML on the prevention of fish balance loss. Dietary EML supplementation offers a feasible way of relieving the stress induced by inappropriately high stocking densities, Cu exposure, and trichlorfon exposure. However, the specific mechanisms need to be further explored.

## Figures and Tables

**Figure 1 animals-13-02652-f001:**
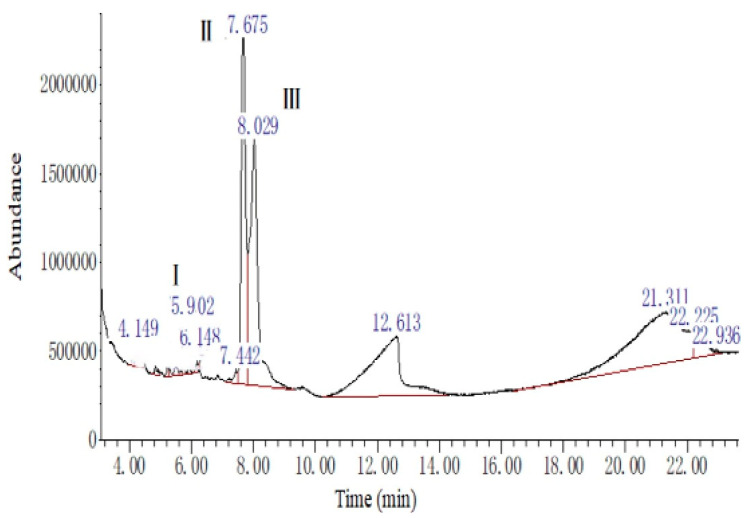
The gas chromatogram of the ethyl ether extract of the mulberry leaf. The numerical value is the retention time. This experiment was repeated three times with similar results achieved.

**Figure 2 animals-13-02652-f002:**
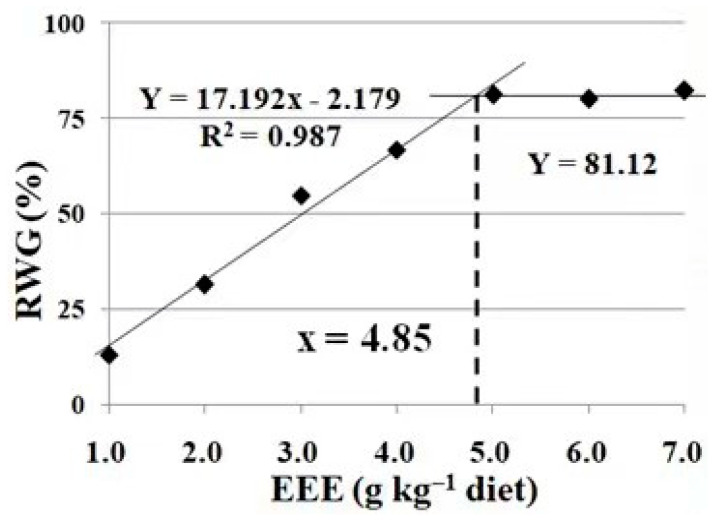
Broken-line analysis of recovery rate of weight gain (RWG) for crucian carp at high stocking densities by feeding diets containing different levels of the mulberry leaf ethyl ether extract (EEE) for 60 days. Values are mean ± SD of 4 replicates.

**Figure 3 animals-13-02652-f003:**
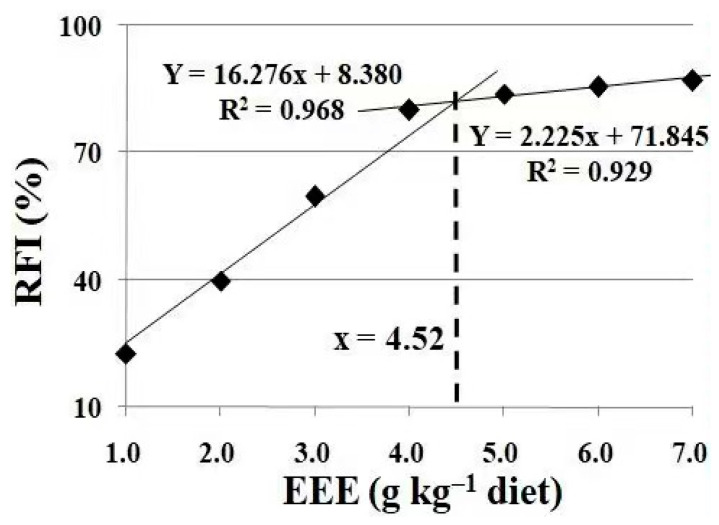
Broken-line analysis of recovery rate of feed intake (RFI) for crucian carp fed diets containing different levels of mulberry leaf ethyl ether extract (EEE) for 30 days, following Cu exposure for 4 days. Values are mean ± SD of three replicates, with 10 fish in each replicate.

**Figure 4 animals-13-02652-f004:**
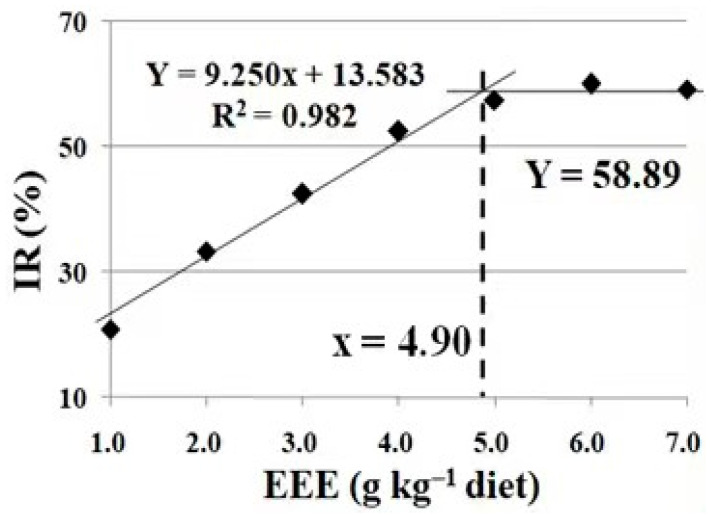
Broken-line analysis of inhibitory rate of rollover (IR) for crucian carp fed diets containing different levels of mulberry leaf ethyl ether extract (EEE) for 30 days, following trichlorfon exposure for 4 days. IR = Y − E (in Table 5). Values are mean ± SD of three replicates, with 10 fish in each replicate.

**Table 1 animals-13-02652-t001:** The composition analyses of the ethyl ether extract of the mulberry leaf using a gas chromatograph–mass spectrometer (GC–MS).

Peak Number	Retention Time (min)	Compound Name	Molecular Weight (amu)	Molecular Formula	Matching Degree (%)
I	5.90	2(4H)-Benzofuranone, 5,6,7,7a-tetrahydro-4,4,7a-trimethyl-	180	C_11_H_16_O_2_	93
II	7.68	Phytol, acetate	338	C_22_H_42_O_2_	83
		Bicyclo [3.1.1]heptane, 2,6,6-trimethyl-, (1.alpha.,2.beta.,5.alpha.)-	138	C_10_H_18_	60
		Bicyclo [3.1.1]heptane, 2,6,6-trimethyl-	138	C_10_H_18_	60
III	8.03	3,7,11,15-Tetramethyl-2-hexadecen-1-ol	296	C_20_H_40_O	87

This experiment was repeated three times, with similar results achieved.

**Table 2 animals-13-02652-t002:** Composition and nutrients content of the basal and experimental diets.

Ingredients	0.00%	0.10%	0.20%	0.30%	0.40%	0.50%	0.60%	0.70%
Fish meal	25.0	25	25	25	25	25	25	25
Soybean meal	32.0	32	32	32	32	32	32	32
Wheat flour	36.6	36.6	36.6	36.6	36.6	36.6	36.6	36.6
DL-methionine	0.70	0.7	0.7	0.7	0.7	0.7	0.7	0.7
Threonine	0.40	0.4	0.4	0.4	0.4	0.4	0.4	0.4
Fish oil	1.50	1.5	1.5	1.5	1.5	1.5	1.5	1.5
Sunflower oil	1.80	1.7	1.6	1.5	1.4	1.3	1.2	1.1
Vitamin mixture ^1^	1.00	1.0	1.0	1.0	1.0	1.0	1.0	1.0
Mineral mixture ^2^	1.00	1.0	1.0	1.0	1.0	1.0	1.0	1.0
EEE	0.00	0.10	0.20	0.30	0.40	0.50	0.60	0.70
Proximate analysis								
Dry matter	92.86	93.12	92.58	93.24	92.67	92.49	93.33	92.45
Crude protein	34.73	34.87	34.69	34.77	34.92	34.86	34.69	34.75
Crude lipid	5.56	5.53	5.57	5.52	5.63	5.54	5.58	5.51
Crude Ash	8.19	8.25	8.31	8.22	8.40	8.24	8.17	8.15

^1^ Per kg of vitamin mix: retinyl acetate (500,000 IU g^−1^), 0.80 g; cholecalciferol (500,000 IU g^−1^), 0.48 g; DL-α-tocopherol acetate (50%), 20.00 g; menadione (23%), 0.43 g; thiamin nitrate (90%), 0.11 g; riboflavine (80%), 0.63 g; pyridoxine HCl (81%), 0.92 g; cyanocobalamin (1%), 0.10 g; ascorhyl acetate (93%), 7.16 g; D-calcium pantothenate (90%), 2.73 g; niacin (99%), 2.82 g; D-biotin (2%), 5.00 g; meso-inositol (99%), 52.33 g; folic acid (96%), 0.52 g. ^2^ Per kg of mineral mix: FeSO_4_·7H_2_O (20% Fe), 69.70 g; CuSO_4_·5H_2_O (25% Cu), 1.20 g; ZnSO_4_·7H_2_O (23% Zn), 21.64 g; MnSO_4_·H_2_O (32% Mn), 4.09 g; Na_2_SeO_3_·5H_2_O (1% Se), 2.50 g; KI (4% I), 2.90 g; CaCO_3_, 897.98 g.

**Table 3 animals-13-02652-t003:** Flavonoid content of cyclohexane extract (CHE), ethyl ether extract (EEE), acetone extract (AE) and aqueous extract (AQE) of mulberry leaf.

Extracts	Flavonoids (mg g Dry Extract^−1^)
CHE	40.11 ± 2.42 ^a^
EEE	63.69 ± 2.01 ^c^
AE	48.65 ± 1.72 ^b^
AQE	41.27 ± 1.65 ^a^

Values are means ± S.D. of 3 replicates. Values in the same column with different superscripts are significantly different (*p* < 0.05).

**Table 4 animals-13-02652-t004:** Initial body weight (IBW), final body weight (FBW), weight gain (WG), specific growth rate (SGR), feed intake (FI), feed efficiency (FE) and survival rate (SR) of crucian carp at high stocking density according to feeding diets containing different levels of the mulberry leaf ethyl ether extract (EEE) for 60 days.

	Densities (Fish L^−1^) + EEE (g kg^−1^ Diet)									Pr > F ^1^		
	0.48 + 0 (K)	0.97 + 0 (Y)	0.97 + 1 (E1)	0.97 + 2 (E2)	0.97 + 3 (E3)	0.97 + 4 (E4)	0.97 + 5 (E5)	0.97 + 6 (E6)	0.97 + 7 (E7)	ANOVA	Linear Trend	Quadratic Trend
IBW (g fish^−1^)	7.20 ± 0.23 ^a^	7.24 ± 0.24 ^a^	7.28 ± 0.26 ^a^	7.23 ± 0.25 ^a^	7.20 ± 0.25 ^a^	7.21 ± 0.26 ^a^	7.26 ± 0.27 ^a^	7.21 ± 0.27 ^a^	7.17 ± 0.28 ^a^			
FBW (g fish^−1^)	34.05 ± 1.35 ^e^	22.25 ± 1.26 ^a^	23.83 ± 1.33 ^a^	25.95 ± 1.12 ^b^	28.69 ± 1.67 ^c^	30.12 ± 0.98 ^cd^	31.88 ± 0.99 ^d^	31.69 ± 1.66 ^d^	31.90 ± 1.37 ^d^	0.00	0.00	0.00
WG (g fish^−1^)	26.85 ± 1.53 ^e^	15.01 ± 1.26 ^a^	16.54 ± 1.50 ^a^	18.72 ± 1.34 ^b^	21.49 ± 1.80 ^c^	22.93 ± 0.87 ^cd^	24.62 ± 1.05 ^d^	24.48 ± 1.52 ^d^	24.73 ± 1.20 ^d^	0.00	0.00	0.00
SGR (% d^−1^)	2.59 ± 0.11 ^e^	1.87 ± 0.10 ^a^	1.97 ± 0.14 ^a^	2.13 ± 0.13 ^b^	2.30 ± 0.13 ^c^	2.39 ± 0.06 ^cd^	2.47 ± 0.08 ^de^	2.47 ± 0.07 ^de^	2.49 ± 0.06 ^de^	0.00	0.00	0.00
FI (g fish^−1^)	39.47 ± 1.80 ^e^	29.95 ± 1.61 ^a^	31.80 ± 1.55 ^a^	33.77 ± 1.16 ^b^	34.85 ± 1.25 ^bc^	36.00 ± 1.09 ^cd^	37.73 ± 1.16 ^de^	37.46 ± 1.15 ^de^	37.11 ± 1.15 ^d^	0.00	0.00	0.00
FE (%)	68.13 ± 5.01 ^b^	50.12 ± 3.49 ^a^	52.03 ± 4.12 ^a^	55.51 ± 4.80 ^a^	61.60 ± 3.49 ^b^	63.74 ± 3.43 ^b^	65.34 ± 4.47 ^b^	65.38 ± 3.75 ^b^	66.76 ± 4.73 ^b^	0.00	0.00	0.00
SR (%)	100.00 ± 0.00 ^a^	100.00 ± 0.00 ^a^	100.00 ± 0.00 ^a^	100.00 ± 0.00 ^a^	100.00 ± 0.00 ^a^	100.00 ± 0.00 ^a^	100.00 ± 0.00 ^a^	100.00 ± 0.00 ^a^	100.00 ± 0.00 ^a^			

Values are mean ± SD of 4 replicates. Values in the same row with different superscripts are significantly different (*p* < 0.05). ^1^ Significance probability associated with the F-statistic.

**Table 5 animals-13-02652-t005:** The activities of trypsin, lipase, amylase, Na^+^/K^+^-ATPase, alkaline phosphatase (AKP), the anti-superoxide anion (ASA), the anti-hydroxy radical (AHR), superoxide dismutase (SOD), catalase (CAT) and glutathione peroxidase (GPx) and the content of malondialdehyde (MDA) in the hepatopancreas or intestine of crucian carp at a high stocking density when feeding them diets containing different levels of the mulberry leaf ethyl ether extract (EEE) for 60 days.

	Densities (Fish L^−1^) + EEE (g kg^−1^ Diet)									Pr > F ^1^		
	0.48 + 0 (K)	0.97 + 0 (Y)	0.97 + 1 (E1)	0.97 + 2 (E2)	0.97 + 3 (E3)	0.97 + 4 (E4)	0.97 + 5 (E5)	0.97 + 6 (E6)	0.97 + 7 (E7)	ANOVA	Linear Trend	Quadratic Trend
Hepatopancreas												
Trypsin (U mg^−1^ protein)	1.02 ± 0.06 ^ab^	0.99 ± 0.06 ^a^	0.99 ± 0.08 ^a^	1.03 ± 0.09 ^ab^	1.15 ± 0.09 ^abc^	1.09 ± 0.11 ^abc^	1.20 ± 0.10 ^c^	1.16 ± 0.09 ^bc^	1.04 ± 0.09 ^abc^	0.04	0.01	0.14
Lipase (U mg^−1^ protein)	43.19 ± 2.67 ^bc^	33.36 ± 2.63 ^a^	37.88 ± 2.52 ^ab^	44.58 ± 2.66 ^c^	43.94 ± 2.72 ^c^	46.27 ± 4.63 ^c^	44.59 ± 4.46 ^c^	45.04 ± 2.69 ^c^	40.95 ± 2.73 ^bc^	0.00	0.01	0.08
ASA (U g^−1^ protein)	60.40 ± 3.89 ^a^	57.55 ± 4.32 ^a^	59.41 ± 4.73 ^a^	61.20 ± 4.65 ^a^	62.60 ± 5.59 ^a^	61.77 ± 4.63 ^a^	60.84 ± 3.06 ^a^	61.20 ± 3.75 ^a^	58.89 ± 5.35 ^a^	0.93	0.68	0.34
MDA (nmol mg^−1^ protein)	10.93 ± 0.80 ^a^	15.86 ± 1.09 ^b^	16.14 ± 1.40 ^b^	15.24 ± 1.18 ^b^	15.38 ± 1.18 ^b^	12.61 ± 0.84 ^a^	11.79 ± 1.03 ^a^	12.03 ± 1.17 ^a^	10.80 ± 1.03 ^a^	0.00	0.00	0.00
CAT (U mg^−1^ protein)	31.91 ± 2.16 ^c^	22.22 ± 1.58 ^a^	22.36 ± 1.95 ^a^	26.18 ± 1.90 ^ab^	27.03 ± 2.03 ^ab^	26.72 ± 2.26 ^ab^	31.05 ± 2.09 ^bc^	33.52 ± 2.37 ^c^	34.77 ± 3.11 ^c^	0.00	0.00	0.00
GPx (U mg^−1^ protein)	442.01 ± 25.59 ^ab^	432.13 ± 28.95 ^a^	458.85 ± 28.95 ^ab^	457.50 ± 36.31 ^ab^	494.25 ± 37.31 ^ab^	505.16 ± 29.45 ^b^	479.75 ± 33.18 ^ab^	481.74 ± 41.72 ^ab^	473.72 ± 33.55 ^ab^	0.02	0.02	0.10
Intestine												
Lipase (U mg^−1^ protein)	42.49 ± 2.63 ^bc^	34.47 ± 2.71 ^a^	39.27 ± 2.72 ^ab^	47.64 ± 2.58 ^cd^	46.15 ± 2.57 ^cd^	48.43 ± 2.47 ^d^	48.99 ± 2.50 ^d^	46.15 ± 2.59 ^cd^	46.69 ± 4.24 ^cd^	0.00	0.00	0.03
Amylase (U mg^−1^ protein)	1.37 ± 0.04 ^d^	1.01 ± 0.04 ^a^	1.19 ± 0.05 ^bc^	1.18 ± 0.08 ^bc^	1.20 ± 0.05 ^bc^	1.24 ± 0.06 ^bc^	1.29 ± 0.07 ^cd^	1.13 ± 0.04 ^b^	1.15 ± 0.06 ^b^	0.00	0.33	0.80
Na^+^/K^+^-ATPase (U mg^−1^ protein)	4.08 ± 0.21 ^b^	3.08 ± 0.21 ^a^	3.44 ± 0.22 ^a^	3.40 ± 0.17 ^a^	3.90 ± 0.22 ^b^	3.86 ± 0.14 ^b^	4.22 ± 0.18 ^b^	3.87 ± 0.14 ^b^	3.92 ± 0.12 ^b^	0.00	0.00	0.10
AKP (U g^−1^ protein)	325.18 ± 19.13 ^b^	261.78 ± 14.06 ^a^	282.11 ± 18.51 ^ab^	278.70 ± 14.61 ^ab^	294.76 ± 19.15 ^ab^	289.99 ± 16.29 ^ab^	295.41 ± 17.97 ^ab^	288.31 ± 19.17 ^ab^	266.22 ± 16.20 ^a^	0.01	0.14	0.67
AHR (U mg^−1^ protein)	163.70 ± 10.54 ^c^	102.81 ± 7.99 ^a^	127.52 ± 10.91 ^b^	122.51 ± 5.74 ^b^	139.07 ± 9.93 ^b^	159.99 ± 11.96 ^c^	177.85 ± 12.72 ^c^	175.49 ± 12.50 ^c^	158.91 ± 9.83 ^c^	0.00	0.00	0.01
MDA (nmol mg^−1^ protein)	13.36 ± 0.74 ^a^	18.71 ± 1.41 ^d^	17.29 ± 0.94 ^cd^	15.94 ± 1.37 ^bc^	16.05 ± 1.29 ^bc^	15.10 ± 1.31 ^abc^	14.39 ± 1.30 ^ab^	16.44 ± 1.19 ^bc^	16.32 ± 1.35 ^bc^	0.00	0.77	0.51
SOD (U mg^−1^ protein)	17.12 ± 1.04 ^a^	18.59 ± 1.00 ^ab^	19.14 ± 1.23 ^ab^	18.94 ± 1.05 ^ab^	19.72 ± 1.11 ^b^	20.16 ± 1.18 ^b^	19.66 ± 1.12 ^b^	19.17 ± 1.25 ^ab^	18.03 ± 0.82 ^ab^	0.03	0.14	0.00
GPx (U mg^−1^ protein)	306.40 ± 22.42 ^ab^	295.78 ± 16.06 ^a^	311.99 ± 24.87 ^ab^	344.56 ± 19.40 ^bc^	339.68 ± 27.76 ^bc^	348.39 ± 28.35 ^bc^	359.49 ± 26.86 ^c^	351.90 ± 21.99 ^bc^	343.71 ± 22.40 ^bc^	0.04	0.00	0.11

Values are mean ± SD of three replicates, with five fish in each replicate. Values with the different superscripts in the same row are significantly different (*p* < 0.05). ^1^ Significance probability associated with the F-statistic.

**Table 6 animals-13-02652-t006:** Feed intake (FI) of crucian carp fed diets containing different levels of mulberry leaf ethyl ether extract (EEE) for 30 days, following Cu exposure for 4 days.

EEE (g kg^−1^ Diet) + Cu (mg L^−1^)	FI (% of Body Weight)
0 + 0.0 (K)	4.49 ± 0.31 ^g^
0 + 0.7 (Y)	0.10 ± 0.00 ^a^
1 + 0.7 (E_1_)	1.09 ± 0.09 ^b^
2 + 0.7 (E_2_)	1.85 ± 0.08 ^c^
3 + 0.7 (E_3_)	2.72 ± 0.17 ^d^
4 + 0.7 (E_4_)	3.61 ± 0.25 ^e^
5 + 0.7 (E_5_)	3.79 ± 0.18 ^ef^
6 + 0.7 (E_6_)	3.85 ± 0.08 ^f^
7 + 0.7 (E_7_)	3.92 ± 0.19 ^f^

Values are mean ± SD of three replicates, with 10 fish in each replicate. Values in the same column with different superscripts are significantly different (*p* < 0.05).

**Table 7 animals-13-02652-t007:** The activities of trypsin, lipase, amylase, alkaline phosphatase (AKP), glutamate–oxaloacetate transaminase (GOT), glutamate–pyruvate transaminase (GPT) in the hepatopancreas or intestine of crucian carp fed diets containing different levels of mulberry leaf ethyl ether extract (EEE) kg^−1^ for 30 days, followed by exposure to 0.7 mg of Cu L^−1^ water for 4 days.

	EEE (g kg^−1^ Diet) + Cu (mg L^−1^)									Pr > F ^1^		
	0.48 + 0 (K)	0.97 + 0 (Y)	0.97 + 1 (E1)	0.97 + 2 (E2)	0.97 + 3 (E3)	0.97 + 4 (E4)	0.97 + 5 (E5)	0.97 + 6 (E6)	0.97 + 7 (E7)	ANOVA	Linear Trend	Quadratic Trend
Hepatopancreas												
Lipase (U mg^−1^ tissue)	1.01 ± 0.07 ^ab^	0.97 ± 0.06 ^a^	0.97 ± 0.07 ^a^	1.02 ± 0.06 ^ab^	1.20 ± 0.09 ^c^	1.14 ± 0.07 ^bc^	1.06 ± 0.09 ^abc^	1.02 ± 0.09 ^ab^	0.98 ± 0.09 ^a^	0.02	0.29	0.01
Amylase (U g^−1^ tissue)	50.10 ± 3.70 ^c^	38.79 ± 2.42 ^a^	43.64 ± 2.41 ^ab^	48.48 ± 2.42 ^bc^	51.72 ± 3.71 ^c^	52.53 ± 3.70 ^c^	50.91 ± 2.43 ^c^	49.29 ± 3.71 ^c^	50.10 ± 2.80 ^c^	0.00	0.00	0.38
GOT (U g^−1^ tissue)	0.81 ± 0.06 ^a^	0.96 ± 0.07 ^b^	0.95 ± 0.06 ^b^	0.96 ± 0.06 ^b^	0.91 ± 0.08 ^ab^	0.87 ± 0.07 ^ab^	0.86 ± 0.07 ^ab^	0.90 ± 0.08 ^ab^	0.95 ± 0.06 ^b^	0.02	0.82	0.42
GPT (U g^−1^ tissue)	0.69 ± 0.04 ^a^	0.94 ± 0.06 ^c^	0.89 ± 0.07 ^bc^	0.86 ± 0.08 ^bc^	0.84 ± 0.06 ^bc^	0.85 ± 0.07 ^bc^	0.78 ± 0.06 ^ab^	0.81 ± 0.07 ^ab^	0.85 ± 0.08 ^bc^	0.02	0.99	0.08
Intestine												
Trypsin (U mg^−1^ protein)	1.50 ± 0.10 ^d^	0.87 ± 0.07 ^a^	0.99 ± 0.06 ^ab^	1.14 ± 0.07 ^bc^	1.64 ± 0.10 ^e^	1.25 ± 0.07 ^c^	1.28 ± 0.10 ^c^	1.29 ± 0.06 ^c^	1.12 ± 0.07 ^bc^	0.00	0.20	0.23
Amylase (U mg^−1^ protein)	1.06 ± 0.02 ^bc^	0.93 ± 0.04 ^a^	0.98 ± 0.03 ^ab^	1.10 ± 0.05 ^c^	1.09 ± 0.02 ^c^	1.23 ± 0.02 ^d^	1.03 ± 0.04 ^bc^	1.04 ± 0.04 ^bc^	0.99 ± 0.04 ^ab^	0.00	0.06	0.00
AKP (U g^−1^ protein)	364.30 ± 25.22 ^cd^	297.42 ± 18.95 ^a^	312.35 ± 13.68 ^ab^	317.40 ± 17.95 ^ab^	352.41 ± 26.68 ^bc^	436.93 ± 30.91 ^e^	422.28 ± 27.12 ^e^	411.41 ± 32.15 ^e^	400.21 ± 25.71 ^de^	0.00	0.00	0.43

Values are mean ± SD of three replicates, with five fish in each replicate. Values with different superscripts in the same row are significantly different (*p* < 0.05). ^1^ Significance probability associated with the F-statistic.

**Table 8 animals-13-02652-t008:** Rollover of crucian carp fed diets containing different levels of mulberry leaf ethyl ether extract (EEE) for 30 days, following trichlorfon exposure for 4 days.

EEE (g kg^−1^ Diet) + Trichlorfon (mg L^−1^)	Rollover (% of Total)
0 + 0.0 (K)	0.00 ± 0.00 ^a^
0 + 2.2 (Y)	100.00 ± 0.00 ^g^
1 + 2.2 (E_1_)	79.17 ± 2.89 ^f^
2 + 2.2 (E_2_)	66.67 ± 5.64 ^e^
3 + 2.2 (E_3_)	57.50 ± 4.61 ^d^
4 + 2.2 (E_4_)	47.50 ± 4.33 ^c^
5 + 2.2 (E_5_)	42.50 ± 2.50 ^bc^
6 + 2.2 (E_6_)	40.00 ± 3.33 ^b^
7 + 2.2 (E_7_)	40.83 ± 3.82 ^b^

Values are mean ± SD of three replicates, with 10 fish in each replicate. Values in the same column with different superscripts are significantly different (*p* < 0.05).

**Table 9 animals-13-02652-t009:** The activities of lactate dehydrogenase (LDH), glutamate–oxaloacetate transaminase (GOT), glutamate–pyruvate transaminase (GPT), the anti-superoxide anion (ASA), the anti-hydroxy radical (AHR), catalase (CAT) and glutathione reductase (GR), and the content of protein carbonyl (PC) in the muscle of crucian carp fed diets containing different levels of mulberry leaf ethyl ether extract (EEE) kg^−1^ for 30 days, followed by exposure to 2.2 mg of trichlorfon L^−1^ water for 4 days.

	EEE (g kg^−1^ Diet) + Trichlorfon (mg L^−1^)									Pr > F ^1^		
	0.48 + 0 (K)	0.97 + 0 (Y)	0.97 + 1 (E1)	0.97 + 2 (E2)	0.97 + 3 (E3)	0.97 + 4 (E4)	0.97 + 5 (E5)	0.97 + 6 (E6)	0.97 + 7 (E7)	ANOVA	Linear Trend	Quadratic Trend
LDH (U mg^−1^ protein)	2.37 ± 0.20 ^c^	1.30 ± 0.10 ^a^	1.43 ± 0.07 ^ab^	1.46 ± 0.13 ^ab^	1.55 ± 0.08 ^b^	1.49 ± 0.08 ^ab^	1.52 ± 0.08 ^ab^	1.44 ± 0.10 ^ab^	1.47 ± 0.12 ^ab^	0.00	0.00	0.00
GOT (U g^−1^ protein)	20.74 ± 1.33 ^b^	16.94 ± 0.96 ^a^	17.46 ± 1.27 ^a^	19.26 ± 0.60 ^ab^	18.88 ± 0.77 ^ab^	19.04 ± 0.78 ^ab^	18.60 ± 0.74 ^ab^	18.51 ± 0.84 ^ab^	18.11 ± 1.19 ^a^	0.01	0.40	0.53
GPT (U g^−1^ protein)	12.07 ± 1.02 ^cd^	8.54 ± 0.53 ^a^	9.67 ± 0.90 ^ab^	9.31 ± 0.69 ^ab^	10.43 ± 0.58 ^bc^	12.82 ± 0.70 ^d^	11.76 ± 0.55 ^cd^	10.43 ± 0.56 ^bc^	10.85 ± 0.72 ^bc^	0.00	0.02	0.41
ASA (U g^−1^ protein)	60.11 ± 4.37 ^ab^	53.23 ± 4.62 ^a^	53.86 ± 3.08 ^a^	53.42 ± 2.77 ^a^	55.24 ± 3.14 ^a^	58.52 ± 3.54 ^ab^	64.12 ± 2.62 ^b^	63.44 ± 4.34 ^b^	63.46 ± 4.53 ^b^	0.00	0.04	0.01
AHR (U mg^−1^ protein)	114.84 ± 2.93 ^d^	79.73 ± 3.38 ^a^	78.90 ± 2.83 ^a^	80.78 ± 2.28 ^a^	83.93 ± 2.64 ^a^	98.33 ± 4.01 ^b^	106.73 ± 3.49 ^c^	109.81 ± 3.15 ^cd^	121.76 ± 3.94 ^e^	0.00	0.00	0.00
PC (nmol mg^−1^ protein)	2.04 ± 0.22 ^ab^	3.05 ± 0.16 ^d^	2.62 ± 0.14 ^c^	2.31 ± 0.14 ^bc^	2.33 ± 0.15 ^bc^	2.07 ± 0.16 ^ab^	2.01 ± 0.16 ^ab^	1.76 ± 0.15 ^a^	1.79 ± 0.13 ^a^	0.00	0.00	0.00
CAT (U mg^−1^ protein)	26.57 ± 2.04 ^c^	16.05 ± 1.00 ^a^	15.94 ± 1.40 ^a^	18.64 ± 1.19 ^ab^	21.23 ± 1.89 ^b^	19.21 ± 1.78 ^ab^	17.83 ± 1.26 ^ab^	18.03 ± 1.55 ^ab^	15.29 ± 1.18 ^a^	0.00	0.00	0.13
GR (U g^−1^ protein)	39.29 ± 1.66 ^c^	23.22 ± 1.55 ^a^	22.76 ± 1.71 ^a^	25.38 ± 1.76 ^a^	27.13 ± 1.68 ^a^	33.52 ± 1.72 ^b^	36.49 ± 3.08 ^bc^	37.58 ± 3.18 ^bc^	36.62 ± 1.81 ^bc^	0.00	0.00	0.00

Values are mean ± SD of three replicates, with five fish in each replicate. Values with different superscripts in the same row are significantly different (*p* < 0.05). ^1^ Significance probability associated with the F-statistic.

## Data Availability

The data used to generate the results in this manuscript can be made available if requested from the corresponding author.

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
