# Peer review of "The Protective Effect of a Dietary Extract of Mulberry (Morus alba L.) Leaves against a High Stocking Density, Copper and Trichlorfon in Crucian Carp (Carassius auratus)"

_animals, 2023, doi:10.3390/ani13162652_

Round 1

Reviewer 1 Report (Previous Reviewer 1)

The authors improved the quality of The MS but still is required much more effort to improve it.

NA

Author Response

thanks for the kind suggestions

Reviewer 2 Report (Previous Reviewer 2)

the tables can be presented vertically, so that all the information can be entered and the results do not overlap

Author Response

Thanks for the kind suggestions. We tried to present the table vertically during the major review, but it's not looks good because of too many row and column. Thanks again for your kind suggestions.

Reviewer 3 Report (Previous Reviewer 3)

now article captioned The protective effect of dietary extract of mulberry (Morus alba  L.) leaves against the high stocking density, copper and tri- chlorfon in crucian carp (Carassius auratus) written by u Chen et al has improved alot. It mees all crietra and standards of the journal so i recommends it acceptance 

Spelling check and gramatical error needs to be fixed

Author Response

thans for the kind suggestions

Reviewer 4 Report (Previous Reviewer 4)

The manuscript presents interesting results and contributes to the advancement of knowledge in this area.

The manuscript has been fully revised, adjusted and it is ready to be published.

Author Response

thanks for the kind suggestions

Reviewer 5 Report (New Reviewer)

In a manuscript submitted for review, the Authors described the protective effect of dietary extract of mulberry leaves against the high stocking density, copper and trichlorfon in crucian carp (Carassius auratus). I find the topic of the manuscript interesting and "up to date" and the whole work is thoughtful. The Authors put a lot of work into preparing this interesting work.

My comments:

1.     there is no clearly defined purpose of the review of the experience, and above all, the conclusion does not answer whether the assumption from the purpose of the research was confirmed.

2. there were 15 fish in the control group and the authors describe this as „optimum", while the other groups had 30 fish each. There is no information what guided the Authors in the number of fish in the groups (whether these are the norms, or the Authors assumed so ...)

3. Mulberry leaf extract was administered at doses ranging from 1.0 to 7.0. g/kg. And here, too, there is no short mention of why the Authors used such doses.

Author Response

This manuscript is a resubmission of an earlier submission. The following is a list of the peer review reports and author responses from that submission.

Round 1

Reviewer 1 Report

The authors investigated the effect of Dietary Extract of Mulberry Leaves on the High Stocking Density, Copper and Trichlorfon Stress in Crucian carp (Carassius auratus). They designed nine treatments to test their hypothesis. This manuscript (MS) was poorly written and not easy to understand. This work could help the sustainability of this species farming if it could be written better. However, some major issues significantly compromised the quality of this MS.

Major comments:

  • First, the manuscript needs to be edited by a native English speaker to improve the language of the MS and fix errors.
  • It is so hard to read and understand this MS. Please avoid using abbreviations as little as possible and use the data with two decimals.
  • The experimental design is hard to understand; why they used only one treatment with different densities, it was 60 days, 30 days with two fish trials, and many more questions that cannot be understood by the MS. They did the first 6 days and move those fish to the new tank? Used the new fish?
  • Why do they expect changes in FI only after 4 days?

·       The statical analysis is required to be revised. If authors want to use both ANOVA and regression should do this for all measured parameters. However, I suggest only focusing on regression. The optimum level based on ANOVA was 75%, but regression was 30%. Only reporting regression can avoid this misconception. The authors should do polynomial regression to see first if the relationship is linear or quadratic. You cannot do quadratic or linear relations with your choice.

The statistical analysis should be changed to regression to see whether there is a polynomial or/and linear regression to eventually provide an optimum level. However, they can report both ANOVA and Regression if it was some parameters that using ANOVA for them make more sense. Please check this paper to see how you can report the results. https://www.sciencedirect.com/science/article/pii/S0044848619305861  

  • I suggest deleting the density effect and easily using regression.
  • Because of these issues, I did not go through line-by-line comments.

Kind regards

NA

Reviewer 2 Report

Table 7 can be modified and presented vertically to be able to use the entire size of the sheet and thus improve the presentation of the results, so the numbers do not overlap

English is generally understood and the ideas are moderately clear, there is no problem in understanding what the author means

Reviewer 3 Report

provided as review report

require extensive gramatical and phrases mistakes. 

Reviewer 4 Report

The manuscript presents interesting results and contributes to the advancement of knowledge in this area.

Some Keywords that already appear in the title can be substituted to improve reader reach. Suggestions are highlighted in the attached text.

Need to format some tables that are highlighted in the attached text.
